# Factors Influencing the Adoption of Magnetic Resonance-Guided High-Intensity Focused Ultrasound for Painful Bone Metastases in Europe, A Group Concept Mapping Study

**DOI:** 10.3390/ijerph20021084

**Published:** 2023-01-07

**Authors:** Julia Simões Corrêa Galendi, Ann-Cathrine Siefen, Debora M. Moretti, Sin Yuin Yeo, Holger Grüll, Grischa Bratke, Alessio Giuseppe Morganti, Alberto Bazzocchi, Chiara Gasperini, Francesca De Felice, Roberto Blanco Sequeiros, Mira Huhtala, Ingrid M. Nijholt, Martijn F. Boomsma, Clemens Bos, Helena M. Verkooijen, Dirk Müller, Stephanie Stock

**Affiliations:** 1Institute for Health Economics and Clinical Epidemiology, Faculty of Medicine and University Hospital of Cologne, University of Cologne, 50935 Cologne, Germany; 2Institute for Food and Resource Economics, Chair for Technology, Innovation Management and Entrepreneurship, University of Bonn, 53115 Bonn, Germany; 3Institute of Diagnostic and Interventional Radiology, Faculty of Medicine and University Hospital of Cologne, University of Cologne, 50937 Cologne, Germany; 4Department of Chemistry, Faculty of Mathematics and Natural Sciences, University of Cologne, 50939 Cologne, Germany; 5Radiation Oncology, IRCCS Azienda Ospedaliero, Universitaria of Bologna, 40138 Bologna, Italy; 6Radiation Oncology, DIMES, Alma Mater Studiorum Bologna University, 40138 Bologna, Italy; 7Diagnostic and Interventional Radiology, IRCCS Istituto Ortopedico Rizzoli, 40136 Bologna, Italy; 8Department of Radiotherapy, Policlinico Umberto I, Sapienza University of Rome, 00161 Rome, Italy; 9Department of Radiology, Turku University Hospital, 20521 Turku, Finland; 10Department of Oncology, Turku University Hospital, University of Turku, 20521 Turku, Finland; 11Department of Radiology, Isala Hospital, 8025 AB Zwolle, The Netherlands; 12Division of Imaging and Oncology, University Medical Center Utrecht, Utrecht University, 3584 CX Utrecht, The Netherlands

**Keywords:** MR-HIFU, bone metastases, cancer pain, implementation science, diffusion of innovation, group concept mapping

## Abstract

Magnetic resonance imaging-guided high-intensity focused ultrasound (MR-HIFU) is an innovative treatment for patients with painful bone metastases. The adoption of MR-HIFU will be influenced by several factors beyond its effectiveness. To identify contextual factors affecting the adoption of MR-HIFU, we conducted a group concept mapping (GCM) study in four European countries. The GCM was conducted in two phases. First, the participants brainstormed statements guided by the focus prompt “One factor that may influence the uptake of MR-HIFU in clinical practice is...”. Second, the participants sorted statements into categories and rated the statements according to their importance and changeability. To generate a concept map, multidimensional scaling and cluster analysis were conducted, and average ratings for each (cluster of) factors were calculated. Forty-five participants contributed to phase I and/or II (56% overall participation rate). The resulting concept map comprises 49 factors, organized in 12 clusters: “competitive treatments”, “physicians’ attitudes”, “alignment of resources”, “logistics and workflow”, “technical disadvantages”, “radiotherapy as first-line therapy”, “aggregating knowledge and improving awareness”, “clinical effectiveness”, “patients’ preferences”, “reimbursement”, “cost-effectiveness” and “hospital costs”. The factors identified echo those from the literature, but their relevance and interrelationship are case-specific. Besides evidence on clinical effectiveness, contextual factors from 10 other clusters should be addressed to support adoption of MR-HIFU.

## 1. Introduction

Pain is a common consequence of bone metastases that substantially reduces the quality of life of patients with advanced cancer [1,2]. For patients with persistent pain despite the use of analgesics, radiotherapy is a well-established treatment option that leads to complete or partial pain relief after two to four weeks in about 60–70% of patients [3,4,5]. Magnetic resonance image-guided high-intensity focused ultrasound (MR-HIFU) is an emerging non-invasive alternative that holds the promise to promote faster pain palliation than radiotherapy in a larger proportion of patients [6,7,8].

HIFU thermally ablates the periosteal nerve and tumor by delivering acoustic energy to the targeted treatment region [9]. HIFU can be performed under the guidance of magnetic resonance imaging (MRI) or ultrasound, but MRI guidance is preferred for bone treatments because MRI thermometry provides a near real-time assessment of temperature and thermal-dose distribution on soft tissues [9]. This enables monitoring the thermal damage on the treated and surrounding healthy tissues, and modulation of the energy level in case the temperature rise is insufficient [9]. MR-HIFU can be performed under general anesthesia or sedation depending on the location of the treatment, the patient characteristics, and the experience of the attending physicians, and it therefore requires an anesthesiologist or sedationist in the MRI room during the procedure [10]. 

After MR-HIFU treatment, pain response occurs within three days, and 67% to 88% of patients have complete or partial pain relief [6,7,11]. To date, no randomized controlled trial (RCT) has been performed to compare the effectiveness of MR-HIFU to radiotherapy. Therefore, a three-armed RCT was designed to compare focused ultrasound and radiotherapy for noninvasive palliative pain treatment in patients with bone metastases—the FURTHER-trial (ClinicalTrials.gov Identifier: NCT04307914).

Evidence from RCTs should underpin the adoption of medical technologies in medical settings, including oncology [12]. However, the adoption of medical technologies encompasses multiple interacting factors, such as the patient’s experience with the underlying illness, the clinician’s resistance to new technologies, the processes of technology application in organizations, financing, and regulatory aspects [13]. These contextual factors have proven to play an even stronger role in the adoption of new technologies than the proof of their effectiveness [12].

Thus, to understand the complexity of the interventions, and the complexity of the social context in which the interventions are being tested, qualitative research is increasingly undertaken alongside RCTs [14]. This is necessary because RCTs may tolerate or control the context, but they do not engage with the context from different perspectives. Moreover, to support the implementation of new technologies, barriers and facilitators from different levels and contexts need to be elicited in order to ground the development of effective implementation strategies [15]. The most common methodologies applied to elicit contextual factors on various levels are focus groups, semi-structured interviews, or mix-method research such as Delphi panels [14]. Group concept mapping (GCM) is one alternative participatory mixed-method research that has been applied to theory development, planning of programs and social interventions, and evaluation of programs in health care [16].

The adoption of MR-HIFU technology is expected to face several challenges, including technical advancements, accumulation of clinical evidence, and reimbursement [17]. However, a systematic evaluation of barriers and facilitators influencing the adoption of MR-HIFU for bone metastases was lacking. To investigate barriers and facilitators influencing the adoption of MR-HIFU in European countries, a GCM approach was applied alongside the FURTHER-trial. Our objective was to elicit the contextual factors influencing the adoption of MR-HIFU, which are not routinely addressed in the RCT design, but could equally impact successful adoption of this technology.

## 2. Materials and Methods

### 2.1. Study Settings

FURTHER is a H2020-funded research project that aims to assess the effectiveness of MR-HIFU to improve early pain palliation for cancer patients with painful bone metastases. The FURTHER project’s main component is a prospective, multicentric, three-arm RCT (ClinicalTrials.gov registration number NCT04307914); it is the first to assess the effectiveness of MR-HIFU compared to either radiotherapy or a combination of MR-HIFU and radiotherapy for pain palliation. Patient recruitment for the trial started on 10.03.2020 in the Netherlands, Germany, Finland, and Italy [18]. The GCM study took place in an early phase of the FURTHER-trial.

### 2.2. GCM 

GCM combines qualitative data obtained from participatory inquiry and multivariate statistical analyses to create concept maps. These concept maps are visual representations that summarize the main ideas of the group (i.e., representing multiple perspectives) and their interrelationships [19,20]. The resulting concept maps express the opinion of the participants on the topic using their own terms and can then be used as a guide for strategically planning the adoption of medical technologies [16,19].

### 2.3. Participant Selection of the GCM Study

The participants represented different perspectives: patients, referring physicians, medical specialists, clinical researchers, technology providers, hospital managers, including members of the FURTHER Consortium. The participants were selected using two different methods. First, purposive sampling was used to ensure diverse representation [21]. Second, snowball sampling (i.e., a chain-referral method) was used to facilitate participant engagement [21].

An invitation letter was sent via email to all identified stakeholders outlining the purpose of the GCM study. The invitation letter included a link to the FURTHER project website, where information on MR-HIFU procedure and the FURTHER project was available. A link was provided at the end of the letter, and those interested in participating created a username and password. A similar invitation was sent before the beginning of phase I and phase II and participation in phase II was independent from phase I.

### 2.4. Data Collection and Analysis

Data collection was conducted online using the platform from Group Wisdom™ (Concept System Inc., Ithaca, NY, USA, Version 2020). At first login, the participants signed electronically an informed consent (provided in Appendix A) and were informed that they could withdraw consent for participation anytime. Participants’ anonymity was guaranteed, and they were asked three to five non-identifying questions about their own background to allow subgroup analyses (File S1).

The GCM study was then conducted in two phases: phase I consisted of a brainstorming task, and phase II comprised sorting and rating tasks. The tasks were conducted in English, with the objective of engaging all countries in creating a single European concept map. Figure 1 summarizes the tasks presented to each participant in each phase and how the data were processed and analyzed.

#### 2.4.1. Phase I—Brainstorming

Phase I took place from 1 August to 31 December 2021. During this period, the participants were asked to brainstorm statements guided by a focus prompt. The focus prompt reflected the research question in a complete-the-sentence format: “One factor that may influence (either positively or negatively) the uptake of MR-HIFU in clinical practice in my country or local context is that...” Reminders were sent by email monthly encouraging the participants to add new statements and to complement the ideas from other participants gathered during that period. Phase I was stopped when the topic was exhausted (i.e., if one week after the last reminder, the participants stopped adding new statements).

To eliminate redundancy and potential ambiguity, the statements added were processed. Two researchers (JSCG and ACS) followed a stepwise approach: (i) splitting statements with more than one idea; (ii) merging redundant statements; (iii) editing the remaining statements to ensure comprehensibility. Finally, one participant revised the resulting list of statements to ensure there were no data loss or changes in meaning.

#### 2.4.2. Phase II—Sorting and Rating

Phase II took place from 12 April to 31 May 2022, and reminders were sent every two weeks. The participants had the choice to log out and resume as many times as needed until the predefined end date of phase II. The statements were presented in a random order for the participants to complete two tasks: sorting and rating the statements.

First, the participants were asked to sort the statements into different piles based on how they consider ideas to be related and to label these piles. The participants were explicitly instructed not to sort statements according to priority or value (e.g., important, hard-to-do) and not to group dissimilar statements into an indefinite pillar (e.g., labeled “other”).

Second, the participants were asked to rate statements on two dimensions: (i) Importance (i.e., how important is this factor for the uptake of MR-HIFU treatment for bone metastases in your country?), and (ii) Changeability (i.e., how possible is it to act on this factor to promote the adoption of MR-HIFU for bone metastases in your country?). To answer both questions, each statement was rated using 5-point Likert scales, from 0 (not at all important/not at all possible) to 4 (extremely important/extremely possible). 

Data generated in phase II were analyzed using the GCM software (Concept System Inc., Version 2020). To generate the point map, multidimensional scaling (MDS) was used to attribute XY coordinates to the statements, which were then plotted into a two-dimensional plane. To understand the cohesiveness between statements, bridging indices were calculated (on a 0 to 1 scale). Bridging indices closer to 0 indicate that a statement was often piled together with statements immediately adjacent to it on the map. Finally, we calculated the stress value for this study. The stress value reflects the discrepancy between the input data matrix (i.e., the original sorting data) and the final point map (Appendix A) [22]. Stress values of previous GCM studies ranged between 0.205 and 0.365. Thus, having a lower stress value than the average of previous studies (0.285) indicates that the participants sorted the statements in a similar manner [19,20].

To develop the cluster map, Ward’s hierarchical cluster analysis was applied to group statements reflecting similar concepts into clusters. To decide on the final number of clusters, two researchers (ACS and JSCG) independently examined several cluster solutions (from 15 to six). Starting with the 15-cluster solution, the clusters were merged one by one until information was lost, which could impact the practicality or interpretability of the cluster map. Bridging indices were considered while constructing the cluster map, and labels derived from the original sort data. A closing session was organized in a hybrid event with all authors to finalize the labeling of the clusters (in cases where a clear preference from the original sort data could not be identified).

Furthermore, we calculated average ratings for each statement and cluster of statements. Average rating values were plotted in pattern matches to show how the clusters were ranked according to importance and changeability. Average ratings were plotted in go-zone displays (i.e., bi-variate graphs for two rating dimensions—importance and changeability). The go-zone is divided into four quadrants (above and below the mean rating for each dimension). Statements falling at the northeast quadrant are important statements, on which it is possible to act, and therefore should be prioritized. The Pearson’s correlation coefficient was calculated to measure the linear relationship between the two rating dimensions. Lastly, subgroup analyses were performed per country, and we calculated the variance of average ratings to determine the coherence between country subgroups.

## 3. Results

### 3.1. Participants

Overall, 79 stakeholders were invited, and 45 of them were involved in at least one phase of this study, resulting in a participation rate of 56%. In phase I, 28 (35%) the participants contributed to the brainstorming task. In phase II, 31 (39%) contributed to the sorting task, 33 (41%) rated statements according to importance, and 29 (36%) according to changeability. Table 1 shows the participants’ characteristics according to each phase of this study.

### 3.2. Collected Statements

Seventy-one statements were collected at the end of phase I. Monthly reminders were useful especially because when the participants logged in a second time, they could read and complement the statements added by other participants. For example, one participant added the statement “reimbursement”; in a second login other participants complemented with the statements “Reimbursement in ambulatory care is essential”, and “Reimbursement is important, both inside the hospital and in ambulatory care”.

After adjusting for redundancy and potential ambiguity, 49 statements entered phase II to be sorted and rated. In the Appendix A detail and exemplify the process of splitting and merging statements.

### 3.3. Concept Maps

Sorting data from 28 participants entered the MDS and cluster analysis. Three participants had to be excluded because they sorted most statements according to priority (e.g., do not agree, important) or value (i.e., two piles of positive vs. negative factors).

The point map (Appendix A) shows the statements (and respective identification numbers) plotted on an x–y chart. The calculated stress value was 0.2560. The cluster map (Figure 2) comprised of 12 clusters: “competitive treatments”, “physicians’ attitudes”, “alignment of resources”, “logistics and workflow”, “technical disadvantages”, “radiotherapy as first-line therapy”, “aggregating knowledge and improving awareness”, “clinical effectiveness”, “patients’ preferences”, “reimbursement”, “cost-effectiveness” and “hospital costs”. Table 2 illustrates one representative statement for each cluster, and a full list of the statements contained in each cluster is provided in the Appendix A.

To ensure internal validity, one adjustment in the clusters had to be made. According to the initial hierarchical cluster analysis, statement 14 (i.e., “difficult patient recruitment, due to a large range in referring medical specialists”) was assigned to the cluster “radiotherapy as first-line therapy”. However, bridging values indicated that statement 14 was often piled with statements from the clusters “physicians’ attitude” and “logistics and workflow”. Because statement 14 matched the issue addressed in the cluster “physicians’ attitude” more appropriately, it was manually moved to this cluster.

#### Importance and Changeability of Statements and Clusters

Pattern matches show the differences between the two rating dimensions (importance vs. changeability) (Figure 3). The cluster “clinical effectiveness” was the most important and the most changeable, while the cluster “competitive treatments” was the least important and the least changeable.

The cluster “clinical effectiveness” was the most important, followed by “radiotherapy as first-line therapy” and “patients’ preferences”. The coherence of perceived importance was notably lower for cluster “reimbursement” and “clinical effectiveness” (i.e., variance between countries 0.34 and 0.14, respectively). Table 3 shows the clusters ranked in order of importance.

For the cluster “reimbursement”, average importance ratings were higher for Germany and the Netherlands (average ratings ≥ 3.00) compared to Italy (average 2.33) and Finland (average 1.83). The low coherence between countries regarding the importance of the cluster “clinical effectiveness” was explained by divergence in one country. Figure 4 shows average ratings on the importance dimension according to country-specific subgroups. In Italy, the most important factors were the availability of anesthesiologists for MR-HIFU procedures (statement 43) and frequency of time slots at the MRI dedicated for HIFU (statement 31), both from the cluster “alignment of resources”.

On the statement level, the factors perceived as most important were: 34—“clinical evidence from randomized clinical trials on the effectiveness of MR-HIFU” (average rating: 3.22); 12—“clear position of HIFU in clinical guidelines” (average rating: 3.18); and 43 —“availability of anesthesiologists for MR-HIFU procedures” (average rating: 3.13). Average ratings for all statements are provided in the Appendix A.

Figure 5 shows average ratings for how important the statements are, and how possible it is to act on each statement to promote the adoption of MR-HIFU. The correlation between the two rating dimensions was high (r = 0.77), resulting in 22 (44%) statements falling at the northeast quadrant (i.e., important statements, on which it is possible to act). Notably, all statements contained in the clusters “clinical effectiveness” and “patients’ preferences” fell into the northeast quadrant. At least one factor from eight other clusters (including “physicians’ attitudes”, “alignment of resources”, “logistics and workflow”, “technical disadvantages”, “radiotherapy as first-line therapy”, “aggregating knowledge and improving awareness”, “reimbursement”, and “cost-effectiveness”) fell into the northeast quadrant.

In contrast, none of the statements from the clusters “competitive treatments” and “hospital costs” fell in the northeast quadrant. Statements located in the northeast quadrant are listed in the Appendix A.

## 4. Discussion

Evidence from the FURTHER-trial is expected to be paramount to the adoption of MR-HIFU but is not enough to ensure successful adoption of this technology. The cluster map developed in our study elicited several individual experiences and offers a conceptual understanding of the factors that may influence the adoption of MR-HIFU in clinical practice. The low stress value (0.25) shows that the participants sorted statements in a similar manner; however, the subgroups per country perceived the importance of these factors slightly differently.

In subgroup analysis per country, reimbursement is notably more important in Germany and the Netherlands compared to Finland, which might be explained by the specific health care financing structures of these countries [23]. For example, in Germany health care providers can negotiate supplementary bundled payment from statutory health insurances for innovative procedures (Neue Untersuchungs- und Behandlungsmethoden) [10]. In contrast, Finland has a system of cost-outlier payment (i.e., individual cases with exceptionally high costs are billed separately) and Finnish municipalities act as both payers and providers of health care [23]. Moreover, in Germany and the Netherlands, the time-lag between collection of data (e.g., resource use) and preparing the data for hospital reimbursement takes in average two years, while in Finland, this time-lag for the data is less than one year [23].

In addition, divergences between countries could be explained by MR-HIFU being at different phases of implementation within the specific organizations or health care systems [24,25]. This could explain why in our results the cluster “clinical effectiveness” is perceived as the most important in all countries, except for Italy where the cluster “alignment of resources” is more important. A multiple case study on the adoption of intensity-modulated radiotherapy found that availability of resources is very important at the pre-implementation phase (i.e., when adopters are still forming an attitude about the innovation). In contrast, clinical evidence becomes more important in the post-implementation phase (i.e., confirming the decision and continuing action) [24].

In health care markets, the adoption of technologies often follows a cyclical and dynamic process, more so for medical devices that are continuously being updated and enhanced with supplementary technology [24]. There are several theories and frameworks describing the diffusion of innovations in health care [25,26]. Based on a literature review of theoretical and empirical studies, Greenhalgh et al. proposed a theoretical framework, the NASSS framework [27]. The NASSS framework stands for Non-adoption, Abandonment, Spread, Scale-up and Sustainability of health and care technologies. According to the NASSS framework, the probability of successful adoption depends on the degree of complexity for seven domains: (i) the condition, (ii) the technology, (iii) the value proposition, (iv) the adopter system, (v) the health care organization and (vi) the wider system, and lastly (vii) the continuous embedding and adaptation over time [13,27].

The statements identified in our study generally fit the domains from the NASSS framework, even though the structure/categorization may deviate in some points [27,28]. For example, the clusters “alignment of resources” and “logistics and workflow” reflect the complexity within the health care organization (domain v), and the cluster “physicians’ attitude” reflects the complexity within the adopter system (domain iv). On the other hand, the statement “bone metastases patients are often unfit for general anesthesia” (ID 45) highlights a complexity that could be intuitively placed within the condition domain. However, this statement was grouped in the cluster “physicians’ attitude” because it was assumed to be an important part of the physicians’ rationale. Hence, although the factors influencing the adoption of MR-HIFU echo previous findings, the relevance of each factor (and how they interact) is notably specific for the case studied [13].

According to our results, to promote adoption of MR-HIFU for pain palliation of bone metastases, clinical evidence from randomized clinical trials (statement 34) is seen as the utmost priority. This might result from the fact that 70% of our participants were involved in the FURTHER-trial. However, previous research has shown that the strength or quality of scientific evidence does not always have a large influence on the decision to adopt innovations in health care [12,29]. For many decision-makers, experiential knowledge can feel more relevant and applicable, and real-world data about the budgetary, operational, and patient impacts can have an equally high impact [12].

Although the cluster “competitive treatments” was perceived as generally unimportant, it is noteworthy that “radiotherapy as first-line treatment” was clustered separately. Radiotherapy is the current standard of care for patients with bone metastases [4], and its importance for the adoption of MR-HIFU is indubitable. However, the competitive advantage of radiotherapy seems difficult to overcome, largely due to the logistic advantages of radiotherapy and the already established referral workflow between care providers.

There were several advantages of GCM alongside a multicentric RCT. First, GCM enables to study the context in which the intervention will be applied, which is normally overlooked by the RCT design. About 30% of the participants were not members of the FURTHER consortium, such as representatives from medical societies and regulatory bodies, who broadened the perspective of an otherwise highly specialized research group. Second, to a multicentric European RCT, the online and asynchronous format was advantageous to engage participants who have busy schedules and are geographically dispersed [30]. Third, GCM brainstorming has been shown to be efficient in terms of time and financial costs compared to other qualitative research approaches such as interviews [31]. Fourth, GCM offered a structured process that allowed engagement of different stakeholders while giving them equal voice and relevance [20]. The anonymous participation in the brainstorming task allowed the participants to respond freely and may offset response behavior that can stem from the hospital hierarchy [20]. Moreover, the involvement of stakeholders in the process itself creates commitment to the adoption of the MR-HIFU [20].

The online GCM format qualified as a reliable and practical solution for stakeholder engagement in the face of the current travel restrictions imposed by the COVID pandemic. However, it should be acknowledged that the COVID pandemic could have influenced the perceived importance of some factors. For instance, the availability of anesthesiologists for MR-HIFU procedures was perceived as a very important factor. Because anesthesiologists were pulled from elective treatments to attend patients with COVID and were broadly unavailable for MR-HIFU treatments, the importance of this factor could have been overestimated.

Because MR-HIFU is in early phase of implementation in clinical practice and the topic is novel, the number of participants was representative to answer the research question. Although GCM studies can have larger sample sizes, the number of participants at each phase was appropriate to perform all the GCM analyses [20]. The overall participation rate was similar to the average participation of online-based qualitative studies, which according to a systematic review is 44.1% [32]. One important limitation of the present GCM study was low patient representation. The patient group consists of older patients with advanced cancer, who have multimorbidity, limited mobility, and limited life expectancy. The online format was thought to be appropriate because it would abstain from in-person interaction (e.g., as needed for focus groups). However, patient recruitment for the FURTHER-trial stopped for two years during the COVID pandemic. As a result, only six patients were invited to participate or to appoint a representative, but five declined due mainly to language barrier. Future studies that intend to apply the GCM methodology in the context of a multinational trial should consider engaging patients in their own language.

## 5. Conclusions

In conclusion, GCM offered a structured process that promoted engagement of different stakeholders alongside the FURTHER-trial. The resulting concept maps shed light on how the participants discern the interrelationships and the relevance of factors that may influence the adoption of MR-HIFU in clinical practice in Europe. Although these are likely to change as the technology evolves and the implementation process continues, the present GCM study was able to construct a common understanding among participants. The findings of this GCM study can be used as a basis to develop strategies and recommendations on how to support the adoption of MR-HIFU in European oncologic care.

## Figures and Tables

**Figure 1 ijerph-20-01084-f001:**
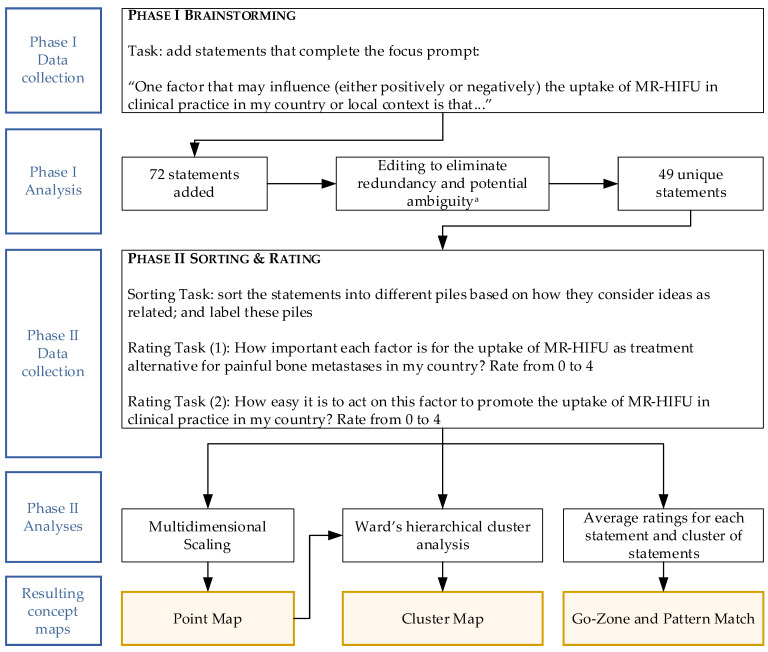
Overview of data collection and analysis for the GCM study. Participants are responsible for generating ideas (phase I) and organizing and structuring the ideas (phase II). ^a^ Performed by two researchers independently.

**Figure 2 ijerph-20-01084-f002:**
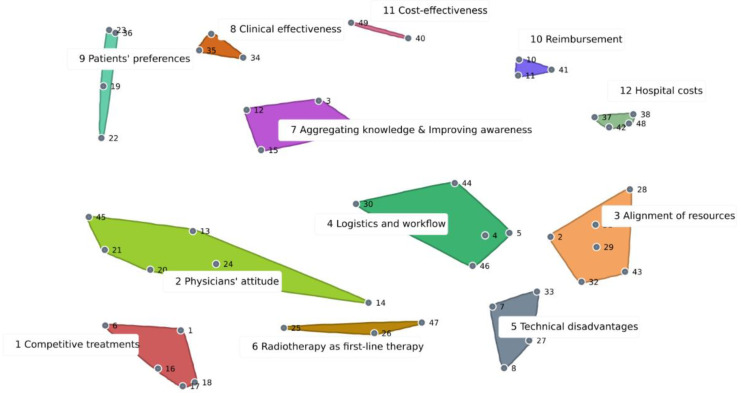
Cluster Map. Statements are numbered and represented by dots. The closer the statements are to each other, the more often they were sorted together by participants.

**Figure 3 ijerph-20-01084-f003:**
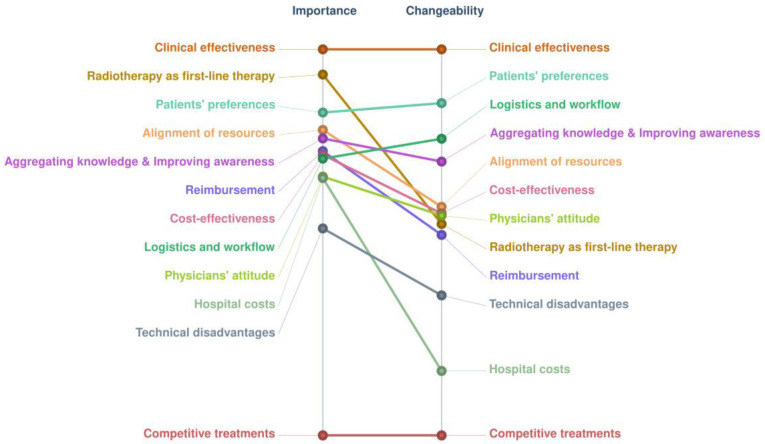
Comparison of the two rating dimensions, importance vs. changeability of clusters. Pattern matches show the average rating value (calculated from Likert scales ranging from 0 to 4), considering results from all participants.

**Figure 4 ijerph-20-01084-f004:**
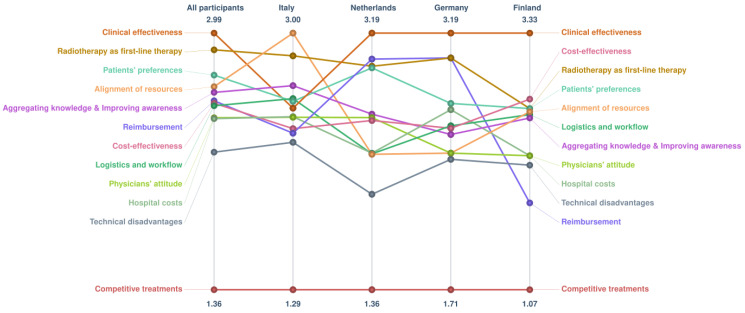
Importance of clusters per country subgroup. Pattern matches show the average rating value (calculated from Likert scales ranging from 0 to 4).

**Figure 5 ijerph-20-01084-f005:**
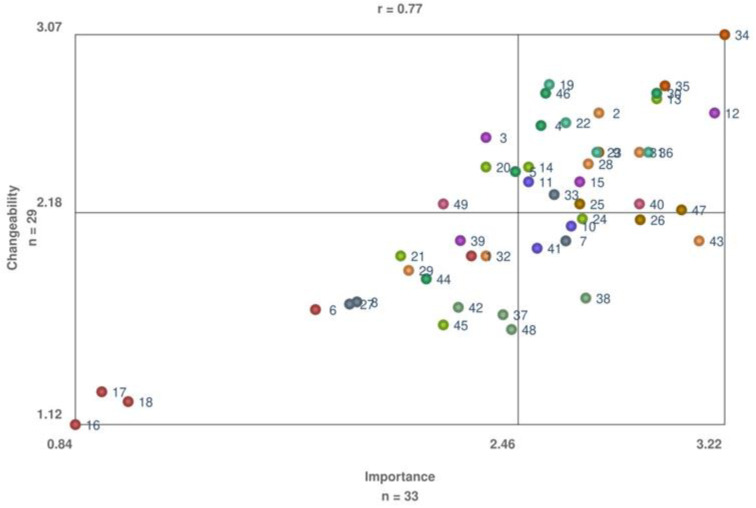
Go-zone display importance vs. changeability (i.e., it is possible to act on these factors to promote the adoption of MR-HIFU). Pearson’s correlation coefficient (r) = 0.77.

**Table 1 ijerph-20-01084-t001:** Participants’ characteristics.

	Phase I	Phase II			All Phases
	Brainstorming	Sorting	Rating Importance	Rating Changeability	All Tasks
Participants	28	31	33	29	45
Member of the FURTHER consortium?					
yes	24 (86%)	24 (71%)	23 (70%)	19 (66%)	32 (71%)
no	4 (14%)	10 (29%)	10 (30%)	10 (34%)	13 (28%)
Per country					
Germany	6 (21%)	7 (21%)	7 (21%)	6 (19%)	9 (20%)
Finland	4 (14%)	6 (18%)	6 (18%)	6 (19%)	7 (16%)
Italy	5 (18%)	11 (32%)	11 (33%)	10 (34%)	12 (27%)
Netherlands	11 (39%)	10 (29%)	9 (27%)	7 (24%)	15 (33%)
Other	2 (7%)	0	0	0	2 (4%)
Expertise in relation to the MR-HIFU provision					
Patient	1 (4%)	0	0	0	1 (4%)
Expertise on performing HIFU treatment	9 (32%)	10 (29%)	10 (30%)	7 (24%)	14 (34%)
Expertise on other medical specialties	9 (32%)	14 (41%)	14 (42%)	14 (48%)	19 (42%)
Expertise on the HIFU technology	7 (25%)	8 (24%)	7 (21%)	6 (21%)	9 (20 %)
Expertise on the Value Proposition/ Financial aspects	2 (7%)	2 (6%)	2 (6%)	2 (7%)	2 (4%)
Self-perceived knowledge on MR-HIFU latest evidence					
Excellent	8 (29%)	6 (18%)	5 (15%)	4 (14%)	9 (20%)
Good	12 (43%)	16 (47%)	16 (48%)	14 (48%)	19 (42%)
Regular	5 (18%)	6 (18%)	6 (18%)	5 (17%)	9 (20%)
Low	2 (7%)	6 (18%)	6 (18%)	6 (21%)	7 (16%)

Abbreviation. MR-HIFU: Magnetic Resonance Image-Guided High-Intensity Focused Ultrasound.

**Table 2 ijerph-20-01084-t002:** Representative statements for each cluster.

Cluster		Statements	
ID Number	Caption	ID Number	Representative Statement (ID)
1	Competitive treatments	6	Availability of ultrasound-guided HIFU as a competitive treatment alternative
2	Physicians’ attitude	13	Unfamiliarity among referring physicians with MR-HIFU as a treatment option
3	Alignment of resources	31	Frequency of time slots at the MRI dedicated for HIFU
4	Logistics and workflow	46	Lack of an established patient workflow (from HIFU-indication to release of the patient)
5	Technical disadvantages	7	MR-HIFU is a lengthy procedure
6	Radiotherapy as first-line therapy	25	HIFU is less flexible with respect to different anatomical regions compared to radiotherapy
7	Aggregating knowledge and Improving awareness	12	Clear position of MR-HIFU in clinical guidelines
8	Clinical effectiveness	34	Clinical evidence from randomized clinical trials on the effectiveness of MR-HIFU
9	Patients’ preferences	19	Enthusiasm for the non-invasive treatment
10	Reimbursement	10	Reimbursement of MR-HIFU as inpatient procedure
11	Cost-effectiveness and	40	Evidence on cost-effectiveness in relation to standard of care
12	Hospital Costs	48	Costs of initial setup (purchase of equipment, installation, etc.)

**Table 3 ijerph-20-01084-t003:** Clusters ranked in order of importance and coherence between countries.

Cluster ID	Cluster	Average Perceived Importance	Coherence of Perception between Countries ^a^
8	Clinical effectiveness	2.99	0.14
6	Radiotherapy as first-line therapy	2.89	0.03
9	Patients’ preferences	2.73	0.03
3	Alignment of resources	2.65	0.08
7	Aggregating knowledge and improving awareness	2.62	0.00
10	Reimbursement	2.56	0.34
11	Cost-effectiveness	2.55	0.02
4	Logistics and workflow	2.53	0.02
2	Physicians’ attitude	2.45	0.02
12	Hospital costs	2.45	0.05
5	Technical disadvantages	2.24	0.03

^a^ Higher variance values reflect lower coherence among countries.

## Data Availability

The data presented in this study are available in the article and Appendix A.

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
