# Peer review of "Factors Influencing the Adoption of Magnetic Resonance-Guided High-Intensity Focused Ultrasound for Painful Bone Metastases in Europe, A Group Concept Mapping Study"

_ijerph, 2023, doi:10.3390/ijerph20021084_

Round 1

Reviewer 1 Report

Thank you for a huge work done for this paper.

However, some considerations for authors:

Paragraph 2.2 If participants are multilingual, from several countries, as in this case, how is the expressive language chosen? As this is explained more in the paragraph 2.4, it would be better to fuse this paragraphs.

Paragraph 2.4.1: As monthly reminders were sent, were they useful? Any more ideas were born? Please describe in this paragraph, in the Figure 1 or elsewhere.

There were a lot of non-participants. Is this a common phenomenom in GSM? Please compare to previous literature.

I think it would be useful to describe a little bit this HIFU techinique, as it is not familiar to all. Maybe also differences to ultrasound-HIFU should be mentioned. Also there is said that lack of anesthesiologist could be barrier to have this method, but it is non-invasive technique. Please explain.

How and with what information the participants received about MRI-HIFU?

To understand the text you have to know a lot of this method. To most of us clinicians this is not familiar. Figure 1 or text should be more informative. The text should be easier to read also to a person who is not in depth familiar with this issue.

As the most of participants were members of the FURTHER consortium, is it possible that the results are biased?

There was no Figure 2, even if there was a figure text for it.

This might be for editors, but Figure 5 and the text for it are separated.

It would be useful to write the meaning of northeast quadrant, etc.

Author Response

Dear reviewers,

Thank you for carefully reviewing the manuscript and for the constructive comments.

We copied the comments below and added responses in red. The changes in the new version of the manuscript are marked with track changes and indicated below. We stay at your disposal for further clarifications.

Reviewer 1

Thank you for a huge work done for this paper.

However, some considerations for authors:

Paragraph 2.2 If participants are multilingual, from several countries, as in this case, how is the expressive language chosen? As this is explained more in the paragraph 2.4, it would be better to fuse this paragraphs.

Thank you for this comment. We changed the phrasing on line 114-115 to clarify the intended meaning.

Paragraph 2.4.1: As monthly reminders were sent, were they useful? Any more ideas were born? Please describe in this paragraph, in the Figure 1 or elsewhere.

Monthly reminders were useful especially because when participants logged in a second time, they could read and complement the statements added by other participants. This process was exhaustive, until no new ideas were added. We added a clarification in the method section in lines 149-150 and in the result section lines 215-220, including an example.

There were a lot of non-participants. Is this a common phenomenom in GSM? Please compare to previous literature.

Thank you for the comment. The pool of stakeholders that are involved with/aware of MR-HIFU is small, but our sample was representative for the research question and enough for GCM analyses. We added this point in the discussion in lines 381-386.

I think it would be useful to describe a little bit this HIFU techinique, as it is not familiar to all. Maybe also differences to ultrasound-HIFU should be mentioned. Also there is said that lack of anesthesiologist could be barrier to have this method, but it is non-invasive technique. Please explain.

We added more information on HIFU technique in the introduction in lines 57-66.

How and with what information the participants received about MRI-HIFU?

Added in lines 125-127.

To understand the text you have to know a lot of this method. To most of us clinicians this is not familiar. Figure 1 or text should be more informative. The text should be easier to read also to a person who is not in depth familiar with this issue.

We rephrased the paragraph 2.2 and added more information on HIFU in the introduction in lines 57-66. We also improved figure 1.

As the most of participants were members of the FURTHER consortium, is it possible that the results are biased?

The results of the group concept mapping study ultimately reflect the opinions of the participants. The possible impacts of the strong participation from the FURTHER Consortium was discussed in lines 346-353.

There was no Figure 2, even if there was a figure text for it.

Figure 2 was added.

This might be for editors, but Figure 5 and the text for it are separated.

Position of figure 5 was corrected.

It would be useful to write the meaning of northeast quadrant, etc.

Meaning added in lines 199-200

Thank you again and we stay at your disposal for further clarifications. 

Reviewer 2 Report

The study applied a Group concept mapping (GCM) alongside the FURTHER-trial to investigate barriers and facilitators influencing the adoption of MR-HIFU in European countries. The authors found that GCM offered a structured process that promoted engagement of different stakeholders alongside the FURTHER-trial. The resulting concept maps shed light on how the participants discerns the interrelationships and relevance of factors that may influence the adoption of MR-HIFU in clinical practice in Europe. The findings of this GCM study can be used as basis to develop strategies and recommendations on how to support the adoption of MR-HIFU in European oncologic care.

   In general, the topic of the study is novel. But the patient representation was low. More patient should be included.

   Detailed evaluation of specific deficiencies with suggestions for improvements:

   1, Line 96. What is the full name of EBRT?

   2, Line 146. “There was no time limit to complete phase II.” So, what is the deadline for this study?

   3,The upper and lower case letters in some part of the article are not corrected. For instance, the words in line 60 are not correct.

Author Response

Dear reviewers,

Thank you for carefully reviewing the manuscript and for the constructive comments.

We copied the comments below and added responses in red. The changes in the new version of the manuscript are marked with track changes and indicated below. We stay at your disposal for further clarifications.

The study applied a Group concept mapping (GCM) alongside the FURTHER-trial to investigate barriers and facilitators influencing the adoption of MR-HIFU in European countries. The authors found that GCM offered a structured process that promoted engagement of different stakeholders alongside the FURTHER-trial. The resulting concept maps shed light on how the participants discerns the interrelationships and relevance of factors that may influence the adoption of MR-HIFU in clinical practice in Europe. The findings of this GCM study can be used as basis to develop strategies and recommendations on how to support the adoption of MR-HIFU in European oncologic care.

   In general, the topic of the study is novel. But the patient representation was low. More patient should be included.

   Detailed evaluation of specific deficiencies with suggestions for improvements:

   1, Line 96. What is the full name of EBRT?

External beam radiotherapy, corrected in lines 105-106.

   2, Line 146. “There was no time limit to complete phase II.” So, what is the deadline for this study?

Phase II took place from April 12th to May 31st, 2022 and participants were aware that at the end date the platform would be closed. We corrected this in the text.

   3,The upper and lower case letters in some part of the article are not corrected. For instance, the words in line 60 are not correct.

We reviewed and corrected the use of upper- and lower-case letters throughout the text.

Thank you again and we stay at your disposal for further clarifications

Round 2

Reviewer 1 Report

Thank you for these corrections. Answers are given well. However, it seems, that there were some lingual inaccuracies after these corrections. For example line 114: This concept maps...; in Figure 1: Phase I and Phase II Data colelction. Please check the whole text again.

Author Response

Dear reviewer, 

Thank you for your revisions and for the positive feedback. 

We conducted a grammar check in the whole text. We used the TrackChanges fucntion to highlight corrections (i.e., grammar check was marked in orange in the new version). 

We stay at your disposal for clarifications, 

Best regards,